# Establishment of an Absolute Quantitative Method to Detect a Plasma tRNA-Derived Fragment and Its Application in the Non-Invasive Diagnosis of Gastric Cancer

**DOI:** 10.3390/ijms24010322

**Published:** 2022-12-24

**Authors:** Xiuchong Yu, Xuemei Song, Yaoyao Xie, Shuangshuang Zhang, Junming Guo

**Affiliations:** Department of Biochemistry and Molecular Biology, Zhejiang Key Laboratory of Pathophysiology, School of Medicine, Ningbo University, Ningbo 315211, China

**Keywords:** tRNA-derived fragment, tRF-27-87R8WP9N1E5, qRT-PCR, gastric cancer, early diagnostics, absolute quantitative detection

## Abstract

(1) Transfer RNA (tRNA)-derived fragments (tRFs) are a new category of regulatory non-coding RNAs with distinct biological functions in cancer. They are produced from pre-tRNAs or mature tRNAs and their sequences are relatively short; thus, the amplification of tRFs, especially those in body fluids, is faced with certain technical difficulties. In this study, we established a quantitative method to detect plasma tRF-27-87R8WP9N1E5 (tRF-27) and used it to screen gastric cancer patients. (2) A specific stem-loop-structure reverse transcription primer, a TaqMan probe, and amplification primers for tRF-27 were prepared, and the absolute quantitative method was used to measure plasma tRF-27 levels. To determine the noninvasive diagnostic value of tRF-27 in gastric cancer, plasma tRF-27 levels in patients with benign and malignant lesions (120 healthy individuals, 48 patients with benign lesions, 48 patients with precancerous lesions, and 72 patients with early gastric cancer) were analyzed. Plasma tRF-27 levels were also analyzed in 106 preoperative gastric cancer patients, 106 postoperative gastric cancer patients, and 120 healthy individuals. Survival curves and Cox regression models were established and analyzed. (3) A new absolute quantitative method to determine the plasma tRF-27 copy number was established. Plasma tRF-27 levels were significantly increased in gastric cancer patients compared to healthy individuals, and the area under the receiver operating characteristic curve was 0.7767, when the cutoff value was 724,807 copies/mL, with sensitivity and specificity values of 0.6226 and 0.8917, respectively. The positive predictive and negative predictive values were 83.50% and 72.80%, respectively. Plasma tRF-27 levels in postoperative gastric cancer patients were significantly decreased compared to preoperative gastric cancer patients and tended to the levels of healthy individuals. Moreover, tRF-27 levels were closely related to tumor size and Ki67 expression in gastric cancer patients. Prognostic analysis showed that tRF-27 may be an independent predictor of overall survival. (4) This novel and non-invasive method of measuring plasma tRF-27 levels was valuable in the early diagnosis of gastric cancer.

## 1. Introduction

Gastric cancer is the fifth-most common type of cancer and the fourth leading cause of cancer-related death in the world [1,2]. Gastric cancer is a molecularly and phenotypically heterogeneous disease that is recently affecting younger populations due to *Helicobacter pylori* infection, poor diet, and other related factors [3,4,5]. Early gastric cancer does not elicit specific symptoms, so many cases of gastric cancer are misdiagnosed or undiagnosed. Therefore, most gastric cancer patients are diagnosed at advanced stages of the disease [6]. If gastric cancer can be detected at an early stage and treated in a timely manner, the 5-year survival rate of patients can reach approximately 90% [7]. Gastroscopy, followed by biopsy and pathology, is currently the gold standard for the diagnosis of this disease, although a certain degree of pain is associated with the procedure [8,9]. Patients’ acceptance of routine blood collection techniques is much higher than that of more invasive methods such as biopsy. Therefore, it is important to identify new serum diagnostic indicators for gastric cancer [10,11,12]. However, commonly used serum biomarkers, such as carcinoembryonic antigen, carbohydrate antigen 19-9, and carbohydrate antigen 72-4, have low sensitivity and specificity, and their detection rate of early gastric cancer is much lower than that of biopsy and pathology [13,14]. Therefore, it is important that diagnostic indicators of gastric cancer be identified, especially those that would result in early diagnosis.

Transfer-RNA-derived fragments (tRFs) are a newly discovered category of regulatory non-coding RNA with value in the diagnosis, treatment, and prognosis of a variety of diseases [15]. tRFs are produced by the enzymatic cleavage of precursor tRNAs or mature tRNAs, and they are 14–30 nucleotides in length [16]. According to the cleavage site, tRFs can be classified into five types, namely, tRF-1, tRF-2, tRF-3, tRF-5, and i-tRF [17]. Furthermore, tRFs may mediate post-transcriptional regulatory events through a mechanism resembling that of microRNAs, in which tRFs bind to RNA-binding proteins or regulate protein translation [18]. Mo et al. reported that the overexpression of tRF-17-79MP9PP in breast cancer cells weakened thrombospondin-1-mediated TGF-β1/SMAD3 signaling, suggesting that it may be a potential therapeutic target for breast cancer [19]. In addition, other studies have demonstrated that the dysregulation of tRFs had clinical significance, and they may be new therapeutic targets [18,20]. Interestingly, previous studies have reported that tRFs were potential diagnostic and prognostic biomarkers of gastric cancer [21], lung cancer [22,23], and colorectal cancer [24]. Therefore, tRFs have unique biological functions that can be exploited to detect different tumor types.

The use of quantitative reverse transcription-polymerase chain reaction (qRT-PCR) to detect gastric-cancer-related RNAs in peripheral blood specimens is a promising technology that can lead to the early detection of diseases [25,26]. Based on our previous study [6], tRF-27-87R8WP9N1E5 (tRF-27) was demonstrated to be a gastric-cancer-associated tRF. tRF-27 is derived from the 5’-terminal positions of mature tRNA-Glu-CTC-2-1 and tRNA-Glu-CTC-1 families, and it belongs to the 5’-tRF type. The “27” in tRF-27-87R8WP9N1E5 represents the number of nucleotides, while “87R8WP9N1E5” indicates the tRF naming system established by Loher and colleagues. The researchers used ten numbers from zero to nine and 22 English alphabet letters, except for A, C, G, and T, to represent the bases [27,28], thereby giving each tRF a unique sequence-based name (https://cm.jefferson.edu/LicensePlates/, accessed on 1 September 2020) and standardizing the uniform nomenclature of tRFs.

In this study, we established an absolute quantitative method to detect plasma tRF-27 levels and analyzed the clinical value of tRF-27 in the early diagnosis of gastric cancer (Figure 1). We believe that this method will improve the non-invasive diagnosis of gastric cancer. 

## 2. Results

### 2.1. Successful Design of tRF-27-Specific Reverse Transcription Primer, TaqMan Probe, and Amplification Primers

The screening of tRF-27 as a gastric cancer-associated tRF is depicted in Figure 2A. The set-up of the optimal PCR reaction system and the program are listed in Appendix A, respectively. The amplification curve showed a flat baseline and an obvious exponential phase, indicating successful amplification of tRF-27 (Figure 2B). The qRT-PCR products were sequenced using T-A cloning. The results were consistent with the sequences of tRF-27, the specific reverse transcription primer, the TaqMan probe, and upstream and downstream amplification primers (Figure 2C), revealing that the primers were designed correctly. 

### 2.2. Establishment of an Absolute Quantitative Method to Detect tRF-27 in the Plasma

The stem-loop-structure containing the tRF-27 sequence was cloned into the pUC57 plasmid to produce the recombinant plasmid pUC57-tRF-27 (Figure 3A). Linear regression analysis was performed, and the tRF-27 standard curve formula was Y = −3.343X + 43.03 (Figure 3B). The correlation coefficient R^2^ was 0.9992, the amplification efficiency E was 0.9913, and the slope was −3.343. According to the standard curve formula and the quantification cycle (*C*_q_) value of qRT-PCR, the absolute amount of tRF-27 in plasma was determined.

### 2.3. Plasma tRF-27 Levels Are Increased in Early Gastric Cancer Patients Compared to Those in Healthy Individuals

The absolute quantitative method was used to determine the tRF-27 copy number in plasma from healthy individuals, as well as patients with benign lesions, preneoplastic lesions, and early gastric cancer. The data are presented as medians (interquartile ranges from 25% to 75%), in units of copies/mL. The healthy individuals had 2.57 × 10^5^ copies/mL (1.76 × 10^5^ copies/mL–4.81 × 10^5^ copies/mL); the patients with benign lesions, 1.35 × 10^6^ copies/mL (5.22 × 10^5^ copies/mL–4.31 × 10^6^ copies/mL); the patients with precancerous lesions, 2.02 × 10^6^ copies/mL (6.43 × 10^5^ copies/mL–4.70 × 10^6^ copies/mL); and the patients with early cancer, 2.15 × 10^6^ copies/mL (5.09 × 10^5^ copies/mL–9.36 × 10^6^ copies/mL). The results showed that plasma tRF-27 levels were increased in patients with early gastric cancer compared to those in healthy individuals (Figure 4).

### 2.4. Plasma tRF-27 Levels Are Significantly Decreased in Postoperative Patients Compared to Preoperative Patients, and Tend to the Level of Healthy Individuals

Plasma tRF-27 levels were measured in the training set and the validation set. In the training set, the 48 patients with preoperative gastric cancer had are 6.48 × 10^5^ copies/mL (2.40 × 10^5^ copies/mL–1.88 × 10^6^ copies/mL); the 48 patients with postoperative gastric cancer, 5.00 × 10^5^ copies/mL (1.24 × 10^5^ copies/mL–1.95 × 10^6^ copies/mL); and the 48 healthy individuals, 2.11 × 10^5^ copies/mL (1.67 × 10^5^ copies/mL–3.12 × 10^5^ copies/mL) (Figure 5A). In the validation set, the 58 patients with preoperative gastric cancer had 2.08 × 10^6^ copies/mL (5.76 × 10^5^ copies/mL–3.62 × 10^6^ copies/mL); the 58 patients with postoperative gastric cancer, 7.44 × 10^5^ copies/mL (1.46 × 10^5^ copies/mL–1.75 × 10^6^ copies/mL); and the 72 healthy individuals, 3.48 × 10^5^ copies/mL (1.95 × 10^5^ copies/mL–5.74 × 10^5^ copies/mL) (Figure 5B). Plasma tRF-27 levels in 71 patients with postoperative gastric cancer were decreased (Figure 5C). The results showed that, compared to healthy individuals, plasma tRF-27 levels were increased in preoperative but decreased in postoperative gastric cancer patients, tending to the level of healthy individuals, which indicates that tRF-27 has potential prognostic value in gastric cancer patients.

### 2.5. tRF-27 Has Good Noninvasive Diagnostic Value in Gastric Cancer

Plasma tRF-27 levels in healthy individuals (*n* = 120) and preoperative gastric cancer patients (*n* = 106) were used to plot the ROC curve. The AUC was 0.7767 (Figure 6A). When the cutoff value was 724,807 copies/mL (Figure 6B), the sensitivity and specificity were 0.6226 and 0.8917, respectively. The positive predictive value and the negative predictive value were 83.50% and 72.80%, respectively. In the preoperative gastric cancer patient group, there were 66 patients with tRF-27 expression levels greater than or equal to 724,807 copies/mL and 40 patients with tRF-27 expression levels lower than 724,807 copies/mL.

### 2.6. Plasma tRF-27 Levels Are Closely Related to Tumor Size and Ki67 Expression in Preoperative Gastric Cancer Patients

The relationship between plasma tRF-27 levels and the clinicopathological factors of preoperative gastric cancer patients was analyzed. The results showed that the plasma tRF-27 level was closely related to the size of tumors and the expression of Ki67, a marker of cell proliferation (Table 1). 

### 2.7. Prognostic Value of tRF-27 in Gastric Cancer Patients

Based on the postoperative follow-up data of gastric cancer patients, we generated an overall survival curve to evaluate the prognosis of patients and observed that the prognosis was poor for patients with high plasma tRF-27 levels compared to patients with low plasma tRF-27 levels (Figure 7). 

### 2.8. Feasibility of tRF-27 as an Independent Predictor of Prognosis

To determine whether the plasma tRF-27 level can influence the overall survival of patients, univariate and multivariate Cox regression models were used, and the results indicated that the plasma tRF27 level was an independent predictor of overall survival (Table 2). 

## 3. Discussion

The involvement of tRFs in the development and progression of cancer is still poorly understood. Our previous study reported that tRF-5026a could inhibit the proliferation, cell cycle progression, and migration of gastric cancer cells by regulating the phosphatase and tensin homologue deleted on chromosome ten/phosphatidylinositol 3-kinase/protein kinase (PTEN/PI3K/AKT) signaling pathway [25]. Furthermore, with the development of high-throughput sequencing, it was found that tRFs exist in various specimens, such as tissues, serum, and plasma [15]. Abnormally expressed tRFs have been demonstrated to be closely related to tumorigenesis, with many tRFs serving as potential biomarkers for cancer diagnosis and prognosis [29,30,31]. Gastric cancer has the highest morbidity and mortality rates in the world [32]. The development of gastric cancer has numerous risk factors such as genetics, gastric ulcer, gastroesophageal reflux disease, infectious agents including Epstein-Barr virus and *Helicobacter pylori* infection, and lifestyle-related risk factors including alcohol and diet [32,33]. Gastric cancer has an asymptomatic nature in the early stage, making timely diagnosis challenging when routine examination including imaging and pathological approaches are used [33]. The key to improving the survival of gastric cancer patients is to diagnose and treat the disease in its early stage [32,33,34]. Blood is the most commonly collected specimen, and the collection of peripheral blood is not invasive and not painful; thus, the procedure is accepted by many individuals [35,36]. In this study, tRF-27 was demonstrated to be a gastric-cancer-associated tRF (Figure 2A). We designed a specific reverse transcription primer and established an absolute quantitative method to measure plasma tRF-27 levels in different groups with gastric cancer to determine its diagnostic and prognostic value (Figure 1). 

We report several innovations in this study. First, we established a new diagnostic technique that optimized and improved the existing reverse transcription procedure. The amplification of tRF-27 has certain technical difficulties for its short sequence. Thus, a specific stem-loop-structure reverse transcription primer was designed to increase the length of tRF-27 for preparing specific cDNA. Then, a TaqMan probe and upstream and downstream amplification primers were designed to enhance the specificity of detection. To verify the specificities of the designed primers, the qRT-PCR products were sequenced using T-A cloning. The sequencing results were consistent with the sequences of tRF-27, the specific reverse transcription primer, the TaqMan probe, and upstream and downstream amplification primers; moreover, the amplification curve showed a flat baseline and an obvious exponential phase, revealing that the primers were designed correctly and successfully amplified tRF-27 (Figure 2B,C). The TaqMan probe could enhance the detection sensitivity, for the fluorescent signal was generated when upstream and downstream primers amplified the sequence bound by the probe, and the signal was derived from the target sequence and not generated by nonspecific amplification. Furthermore, by establishing a standard curve for interpolation, an absolute quantitative method for the measurement of plasma tRF-27 levels was established (Figure 3), which resolved the problem of no reliable internal reference to quantify RNA molecules in bodily fluids. The absolute quantitative method can be used to calculate the tRF-27 copy number in plasma specimens of different groups. 

Second, we proposed a new diagnostic and prognostic biomarker for gastric cancer. Gastric cancer usually has a poor prognosis because it is generally diagnosed at an advanced stage [35]. The occurrence of gastric cancer is a multi-stage and gradual development process, from benign lesions (chronic superficial gastritis), precancerous lesions (chronic atrophic gastritis with intestinal metaplasia, atypical hyperplasia of gastric mucosa), and early gastric cancer to advanced gastric cancer [37]. As a result, understanding the process of gastric cancer is critical to identify the molecular biomarkers for early diagnosis. By analyzing plasma tRF-27 levels in patients with benign to malignant lesions (healthy individuals, as well as patients with benign lesions, precancerous lesions, or early gastric cancer), we observed that plasma tRF-27 levels were significantly increased in early gastric cancer patients compared to healthy individuals (Figure 4). In addition, by comparing plasma tRF-27 levels among preoperative patients, postoperative patients, and healthy individuals, we observed that plasma tRF-27 levels decreased in most postoperative gastric cancer patients, tending to the levels of healthy individuals (Figure 5). These results indicate that tRF-27 has potential diagnostic and prognostic value. The ROC curve showed that tRF-27 had high value in the diagnosis of gastric cancer. The AUC was 0.7767. When the cutoff value was 724,807 copies/mL, the sensitivity and the specificity were 0.6226 and 0.8917, respectively. The positive predictive value and the negative predictive value were 83.50% and 72.80%, respectively (Figure 6). Plasma tRF-27 levels were also closely related to tumor size and Ki67 expression in preoperative gastric cancer patients (Table 1). This indicates that tRF-27 levels may be closely related to the prognosis. More importantly, prognostic analysis showed that patients with higher plasma tRF-27 levels had poor prognoses (Figure 7). Univariate and multivariate Cox regression analysis showed that the plasma tRF27 level was an independent predictor of overall survival (Table 2). Thus, tRF27 has good non-invasive diagnostic and prognostic value in gastric cancer. 

Our study had some limitations. This is the first study, and additional analyses should be performed on larger populations in the future. In addition, the method should be validated by different laboratories.

## 4. Materials and Methods

### 4.1. Design of a Specific Stem-Loop-Structure Reverse Transcription Primer of tRF-27

The sequence of tRF-27 is relatively short at only 27 nucleotides. Thus, its amplification is associated with certain technical difficulties. We designed a specific stem-loop-structure reverse transcription primer to increase the length of tRF-27 for preparing cDNA (Appendix A). The sequence of the reverse transcription primer was 5′–GCAGACGAGGGTACCTCCTCTCTTCTCTACTCGTGTCCTACCCTCGTCTGCGAATCC–3′. The optimal concentration of the reverse transcription primer was 0.005 μM, that is, the amount added to the system was 2 μL. The total RNA amount added to the system was 6 μL. The reverse transcription reaction components were prepared as listed in Appendix A. The First strand cDNA Synthesis Kit (with gDNA removal) (Tiosbio, Beijing, China) was used for the preparation of cDNA.

We optimized the existing reverse transcription steps and developed a two-step reverse transcription program (Appendix A). The cDNA was stored at −20 °C or used immediately for qRT-PCR.

### 4.2. Design of a TaqMan Probe and Amplification Primers for qRT-PCR Detection 

The preparation of the TaqMan probe was specific and accurate. The sequence of the forward primer was 5′–GGTCTCTGGTGGTCTAGTGGT–3′, the sequence of the reverse primer was 5′–GGTACCTCCTCTCTTCTCTACT–3′, and the sequence of the TaqMan probe was 5′–FAM–AGGATTCGCAGACGAGGGTAGGACA–TAMRA–3′. The optimal concentration of the upstream primer, downstream primer, and TaqMan probe was 10 μM, that is, the amounts added to the system were 1.4 μL, 1.4 μL, and 0.8 μL, respectively. The cDNA amount added to the system was 0.8 μL. The set-up of the TaqMan probe reaction system and the program are listed in Appendix A, respectively. The qRT-PCR products were sequenced using T-A cloning to verify the specificities of the designed primers.

### 4.3. Establishment of an Absolute Quantitative Method to Measure Plasma tRF-27 Levels

There is no reliable method to quantify RNA molecules in bodily fluids. In this study, we established an absolute quantitative method to quantify plasma tRF-27 levels. The stem-loop-structure containing the tRF-27 sequence was cloned into the pUC57 plasmid (General Biology, Hefei, China) to prepare the recombinant plasmid pUC57-tRF-27. To prepare high-concentration standards, the lyophilized plasmid was dissolved in 50 μL of DNase-free water, and eight dilution gradients were prepared for qRT-PCR. The Applied Biosystems™ QuantStudio™ 3 Quantitative Real-Time PCR System (Thermo Fisher Scientific, Waltham, MA, USA) coupled to QuantStudio™ Design and Analysis Software was used to detect the quantification cycle (*C*_q_) value. Based on the copy number of the template and the *C*_q_ value of qRT-PCR, a standard curve of the tRF-27 copy number was generated. According to the *C*_q_ value and the standard curve formula, the absolute amount of tRF-27 in plasma was calculated.

### 4.4. Detection of tRF-27 Levels in Patients with Benign and Malignant Gastric Lesions

Peripheral venous blood was collected into ethylenediamine tetraacetic acid anticoagulation tubes and centrifuged for 10 min at 1500× *g* and 4 °C. The upper layer, that is, the plasma (250 μL), was transferred into Eppendorf tubes for the extraction of total RNA with TRIzol LS (Invitrogen, Carlsbad, CA, USA). RNA was dissolved in 7 μL of RNase-free water, and 1 μL of each sample was used for the determination of the purity and concentration with a Nanodrop UV Spectrophotometer (Denovix, Houston, TX, USA). If the ratio of A260 to 280 was between 1.8 and 2.1, the RNA purity met the requirements for further experimentation. Next, the remaining 6 μL of each sample was used for reverse transcription. Peripheral venous blood from 120 healthy individuals, 48 patients with benign lesions, 48 patients with precancerous lesions (chronic atrophic gastritis with intestinal metaplasia or gastric mucosal dysplasia), and 72 patients with early gastric cancer were obtained from the Department of Gastroenterology, Affiliated Hospital of the Medical School of Ningbo University (Ningbo, China). The absolute quantitation method was used to measure plasma tRF-27 levels in patients with benign and malignant gastric lesions.

### 4.5. Investigation of Training and Validation Sets

Plasma tRF-27 levels were measured in the training set (48 pairs of preoperative and postoperative plasma samples from gastric cancer patients, and plasma samples from 48 healthy individuals) and the validation set (58 pairs of preoperative and postoperative plasma samples from gastric cancer patients, and plasma samples from 72 healthy individuals). The preoperative and postoperative plasma samples of gastric cancer patients were obtained from the Department of Gastrointestinal Surgery, Ningbo First Hospital (Ningbo, China). The plasma samples of healthy individuals were obtained from the Department of Gastroenterology, Affiliated Hospital of the Medical School of Ningbo University. 

### 4.6. Construction of a Receiver Operating Characteristic Curve to Evaluate the Diagnostic Value of tRF-27 in Gastric Cancer

Plasma tRF-27 levels in 120 healthy individuals and 106 patients with preoperative gastric cancer were measured. The receiver operating characteristic (ROC) curve was used to evaluate the diagnostic value. The area under the curve (AUC), cutoff value, sensitivity, specificity, positive predictive value, and negative predictive value were analyzed using GraphPad Prism 9 Software (version 9.0.0, San Diego, CA, USA). The ROC curve was drawn with 1-specificity as the abscissa and sensitivity as the ordinate. The size of the AUC can quantitatively and specifically indicate the accuracy of the diagnostic test. The criterion of the cutoff value was set up according to Youden’s index. Youden’s index is a method to evaluate the authenticity of the screening test. Youden’s index was calculated as follows: (sensitivity + specificity) − 1, and the larger the index, the better the effect and the authenticity of the screening test. The positive predictive value was calculated as follows: true positive/(true positive + false positive). The negative predictive value was calculated as follows: true negative/(true negative + false negative).

### 4.7. Analysis of Clinicopathological Data and Construction of a Survival Curve

The relationship between plasma tRF-27 levels and clinicopathological factors was analyzed. Combined with the follow-up data, a survival curve was generated to analyze the potential of tRF-27 as a prognostic marker for gastric cancer. Univariate and multivariate Cox regression models were used to determine whether the plasma tRF-27 level is an independent predictor of overall survival.

### 4.8. Statistical Analysis

Statistical analysis was performed with Statistical Product and Service Solutions (SPSS) Software (version 23.0, Chicago, IL, USA). Graphs were plotted with GraphPad Prism 9 Software. For pairwise comparisons, *χ*^2^ tests or Mann–Whitney *U* tests were used according to the different situations. Multiple groups were compared using Kruskal–Wallis *H* tests when the absolute quantification in plasma did not conform to a normal distribution. Graphs were presented with medians and interquartile ranges. The standard curve of absolute quantitation was generated with linear regression analysis. Correlations were determined using Spearman correlation analysis. The log-rank (Mantel–Cox) test was used for survival analysis. Prognostic risk prediction was performed using univariate and multivariate Cox regression models. A *p*-value < 0.05 indicated a statistically significant difference.

## 5. Conclusions

In conclusion, this study establishes an absolute quantitative method to measure plasma tRF levels and applies the method to screen gastric cancer patients. This method was useful to determine the tRF-27 copy number in plasma specimens from patients with gastric cancer. Therefore, this method may improve the diagnosis of patients with gastric cancer.

## Figures and Tables

**Figure 1 ijms-24-00322-f001:**
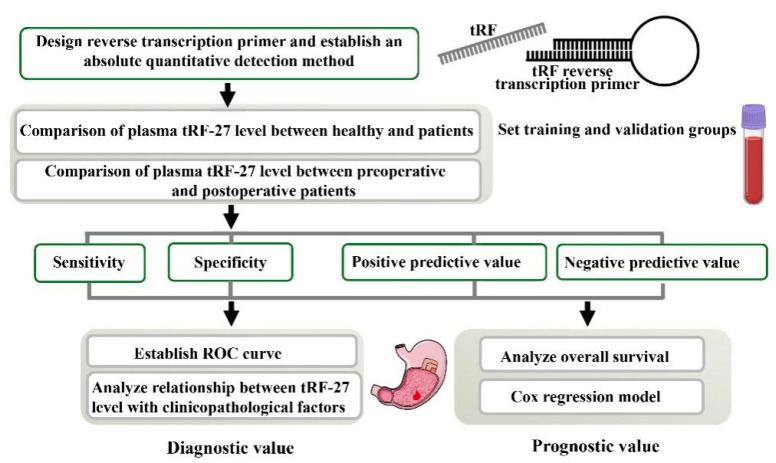
Flow chart of the study design. ROC, receiver operating characteristic.

**Figure 2 ijms-24-00322-f002:**
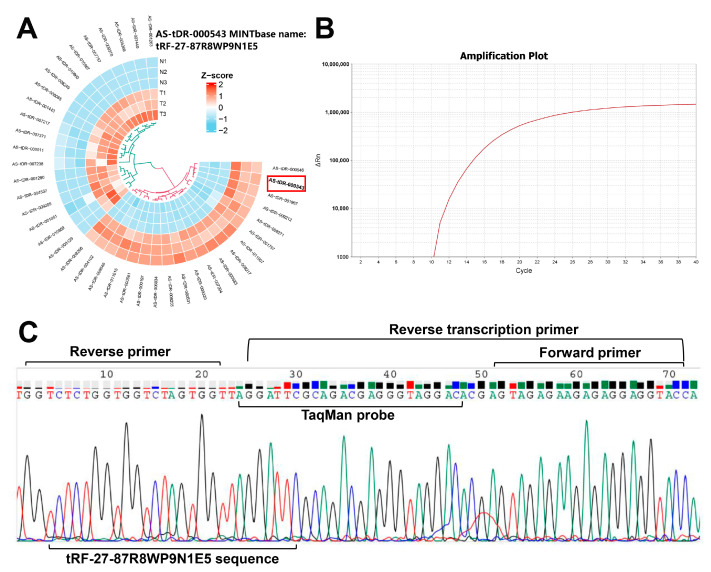
Amplification curve and design of qRT-PCR primers for tRF-27. (**A**) Screening of tRF-27 as a gastric cancer-related tRF. (**B**) Amplification curve. (**C**) Results of T-A cloning of qRT-PCR products.

**Figure 3 ijms-24-00322-f003:**
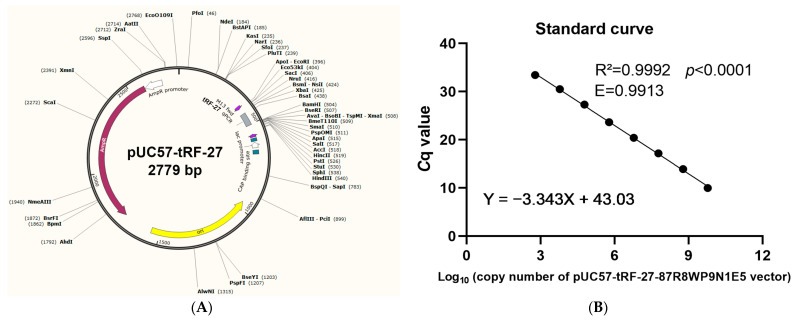
Standard curve used for the absolute quantification of plasma tRF-27 levels. (**A**) Map of the pUC57-tRF-27 recombinant plasmid. (**B**) Standard curve of absolute quantitation.

**Figure 4 ijms-24-00322-f004:**
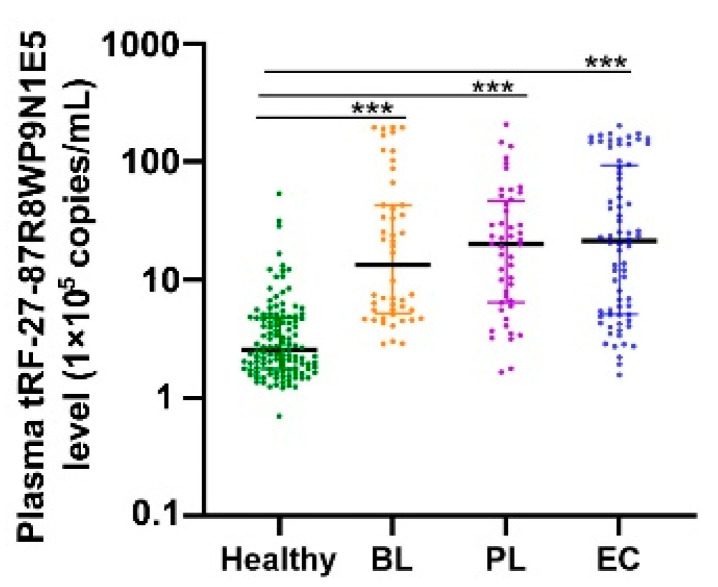
Plasma tRF-27 levels in healthy individuals, patients with benign gastric lesions, and patients with malignant gastric lesions. Data are presented as medians (interquartile ranges from 25% to 75%): healthy individuals (*n* = 120); BL, patients with benign lesions (*n* = 48); PL, patients with precancerous lesions (*n* = 48); and EC, patients with early cancer (*n* = 72). The colors indicate different groups. *** *p* < 0.001.

**Figure 5 ijms-24-00322-f005:**
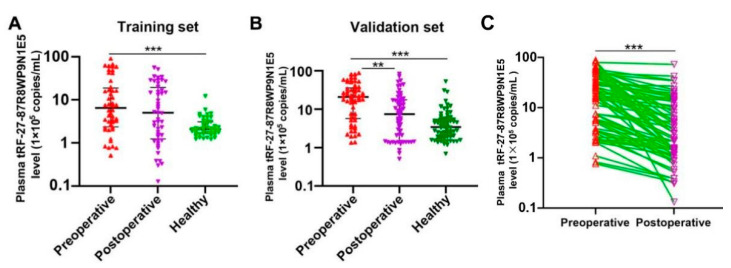
Plasma tRF-27 levels in patients with preoperative and postoperative gastric cancer and healthy individuals. Data are presented as medians (interquartile ranges from 25% to 75%). (**A**) Training set. Patients with preoperative gastric cancer (*n* = 48); patients with postoperative gastric cancer (*n* = 48); and healthy individuals (*n* = 48). (**B**) Validation set. Patients with preoperative gastric cancer (*n* = 58); patients with postoperative gastric cancer (*n* = 58); and healthy individuals (*n* = 72). (**C**) Plasma tRF-27 levels in postoperative patients (*n* = 71) were decreased. The colors indicate different groups. ** *p* < 0.01; *** *p* < 0.001.

**Figure 6 ijms-24-00322-f006:**
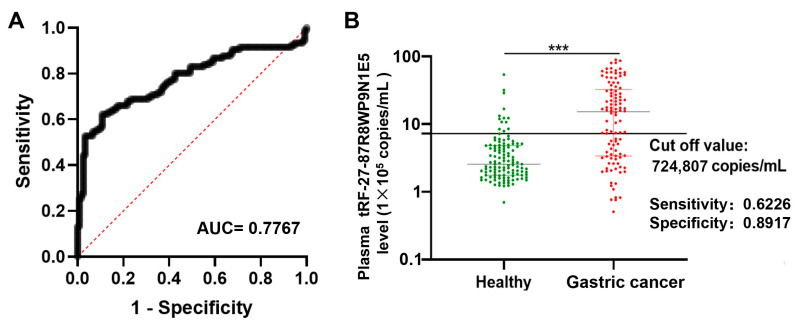
Diagnostic value of tRF-27 in gastric cancer. (**A**) ROC curve of tRF-27 in gastric cancer. (**B**) Cutoff value of tRF-27 in gastric cancer patients was 724,807 copies/mL. Healthy individuals (*n* = 120); Gastric cancer (*n* = 106). *** *p* < 0.001.

**Figure 7 ijms-24-00322-f007:**
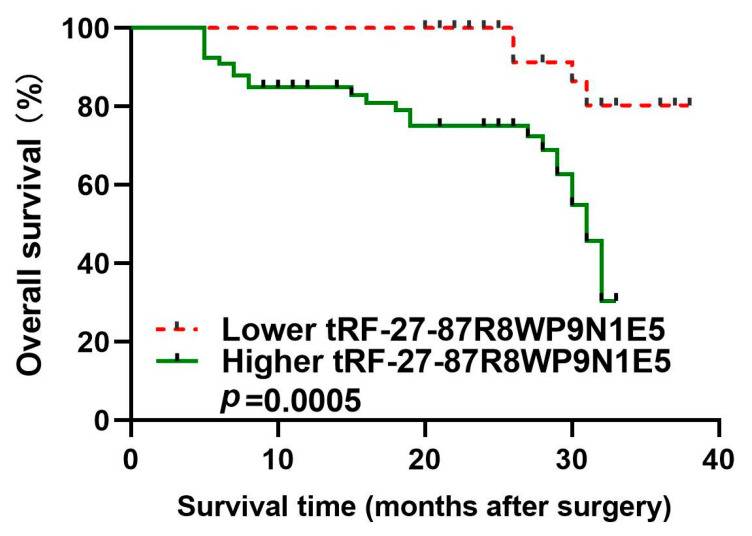
Prognostic value of tRF-27 in gastric cancer patients. High tRF group (*n* = 66); low tRF group (*n* = 40). High tRF group: tRF-27 expression was greater than or equal to 724,807 copies/mL. Low tRF group: tRF-27 expression level was lower than 724,807 copies/mL.

**Table 1 ijms-24-00322-t001:** Relationship between plasma tRF-27 levels and the clinicopathological factors of preoperative gastric cancer patients.

Characteristic	*n* (%)	High (%)	Low (%)	*p*-Value
All cases	106 (100)	66 (62.3)	40 (37.7)	
Gender				0.987
Male	69 (65.1)	43 (65.2)	26 (65.0)	
Female	37 (34.9)	23 (34.8)	14 (35.0)	
Age (y)				0.758
<60	31 (29.2)	20 (30.3)	11 (27.5)	
≥60	75 (70.8)	46 (69.7)	29 (72.5)	
Tumor size (cm)				0.026
<5	71 (67.0)	39 (59.1)	32 (80.0)	
≥5	35 (33.0)	27 (40.9)	8 (20.0)	
Differentiation				0.671
Poor	77 (72.6)	47 (71.2)	30 (75.0)	
Moderate–Well	29 (27.4)	19 (28.8)	10 (25.0)	
TNM stage				0.402
0 and I	35 (33.0)	21 (31.8)	14 (35.0)	
II	22 (20.8)	17 (25.8)	5 (12.5)	
II	41 (38.7)	22 (33.3)	19 (47.5)	
IV	8 (7.5)	6 (9.1)	2 (5.0)	
T stage				0.191
Tis and T1	29 (27.4)	17 (25.8)	12 (30.0)	
T2 and T3	20 (18.9)	16 (24.2)	4 (10.0)	
T4	57 (53.7)	33 (50.0)	24 (60.0)	
Lymphatic metastasis				0.691
N0 and N1	61 (57.5)	37 (56.1)	24 (60.0)	
N2 and N3	45 (42.5)	29 (43.9)	16 (40.0)	
Distal metastasis				0.439
M0	98 (92.4)	60 (90.9)	38 (95.0)	
M1	8 (7.6)	6 (9.1)	2 (5.0)	
CEA				0.845
Negative	91 (85.8)	57 (86.4)	34 (85.0)	
Positive	15 (14.2)	9 (13.6)	6 (15.0)	
CA19-9				0.339
Negative	91 (85.8)	55 (83.3)	36 (90.0)	
Positive	15 (14.2)	11 (16.7)	4 (10.0)	
Ki 67 (%)				0.005
<50	38 (35.8)	17 (25.8)	21 (52.5)	
≥50	68 (64.2)	49 (74.2)	19 (47.5)	

High, tRF-27 expression was greater than or equal to 724,807 copies/mL; Low, tRF-27 expression was lower than 724,807 copies/mL; CEA, carcinoembryonic antigen; CA19-9, carbohydrate antigen 19-9.

**Table 2 ijms-24-00322-t002:** Plasma tRF-27 is an independent predictor of overall survival of gastric cancer patients.

Variable	HR	95% CI	*p*-Value
Univariate analysis			
Gender (Male vs. Female)	0.542	0.216–1.362	0.193
Age (<60 y vs. ≥60 y)	1.120	0.483–2.598	0.791
Tumor size (<5 cm vs. ≥5 cm)	0.508	0.230–1.122	0.094
Differentiation (Poor vs. Moderate–Well)	0.677	0.269–1.700	0.406
TNM stage (0 and I vs. II vs. III vs. IV)	1.442	0.969–2.146	0.071
Invasion depth (Tis and T1 vs. T2 and T3 vs. T4)	1.587	0.956–2.637	0.074
Lymphatic metastasis (N0 and N1 vs. N2 and N3)	0.455	0.204–1.013	0.054
Distal metastasis (M0 vs. M1)	0.557	0.166–1.864	0.342
CEA (Negative vs. Positive)	1.037	0.354–3.039	0.947
CA19-9 (Negative vs. Positive)	0.430	0.168–1.103	0.079
Ki 67 (<50% vs. ≥50%)	0.491	0.196–1.230	0.129
Expression of tRF-27 in plasma (High vs. Low)	0.175	0.058–0.525	0.002
Multivariate analysis			
Expression of tRF-27 in plasma (High vs. Low)	0.144	0.042–0.491	0.002

HR, hazard ratio; CI, confidence interval.

## Data Availability

The datasets used and/or analyzed in the current study are available from the corresponding author upon reasonable request.

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
