# Peer review of "Establishment of an Absolute Quantitative Method to Detect a Plasma tRNA-Derived Fragment and Its Application in the Non-Invasive Diagnosis of Gastric Cancer"

_ijms, 2022, doi:10.3390/ijms24010322_

Round 1

Reviewer 1 Report

Dear Authors,

The manuscript submitted by Yu,et al describes a qPCR based method to detect a certain tRNA in plasma and further investigates it clinical value in early detection of gastric cancer, providing a reasonably large cohort of patient data. However the presentation of data is poor and scientific writing in English need largely improved.

Please amend the manuscript by addressing the following comments:

1. The description on miRNA in the first paragraph is not very related the context. Authors could introduce this later or in discussion part.

2.Auggest to include more information on tRFs, specially how they are regulating cell signaling pathway in second or third paragraph.

3. Figure 1 should be more concise

4. In the method and result part, the authors seem to mix up when describing. For example Table 1 is mentioned in method part, but results are discussed very late in result part. I would suggest authors to re-write them all while keeping concise. Also author could consider to put assay set up in the main figures instead of supplemental figure. 

5. For result part, authors should describe patient clearly (for example, 106 patients in table1 are not clear, are they from same cohort?).

6. Explain results in more details: in result 3.1 and others, Please describe data by mentioning the average, mean, etc. Same with figure legend, author should clearly describe statistics terminology

7. In Figure 5, what is the cutoff for low and high expression of tRF-27? what is the criterion to set up in this way?

8. Authors should their research (background, findings, significance) in a broader way in discussion part

I suggest the manuscript is under major revision.  

Author Response

Ref.: Ms. No. ijms-2033529

Title: "Establishment of an absolute quantitative method to detect a plasma tRNA-derived fragment and its application in the non-invasive diagnosis of gastric cancer"

Dear Reviewer,

On behalf of my co-authors, we thank you very much for giving us an opportunity to revise our manuscript entitled “Establishment of an absolute quantitative method to detect a plasma tRNA-derived fragment and its application in the non-invasive diagnosis of gastric cancer” (Manuscript ID: ijms-2033529). We appreciate the editors and reviewers very much for your positive and constructive comments. Those comments are all valuable and immensely helpful for improving our paper.    

We have studied all peer reviewers’ comments carefully and have addressed the suggestions raised by reviewers and editors. You endorsed this manuscript and we carefully revised the paper. More data and discussion have been added. We have submitted our documents with tracked changes to highlight the revisions. Point-to-point responses to the associate editors’ and reviewers’ comments are listed below this letter. Attached please find the revised version, which we would like to submit for your kind consideration.    

This manuscript was suitable for the special topic “Epigenetic Biomarkers and Applications for Liquid Biopsy Based Diagnostics”. We are willing to address your concerns carefully to make this manuscript suitable for publication, for the first author Xiuchong Yu is expected to graduate from the doctoral program in June 2023. In order to successfully obtain the doctoral degree, this paper needs to be published before March 2023. Every step forward, we are full of gladness and expectation.

We would like to express our great appreciation to you and reviewers for comments on our paper. Looking forward to hearing from you.

With best wishes,

Yours sincerely,

Junming Guo

Director and Professor

Department of Biochemistry and Molecular Biology,

and Zhejiang Key Laboratory of Pathophysiology,

School of Medicine, Ningbo University,

Ningbo 315211,

China

Tel: +86-574-87600756; Fax: +86-574-87608638;

Respond to Reviewer 1

General Comments: Dear Authors, the manuscript submitted by Yu,et al describes a qPCR based method to detect a certain tRNA in plasma and further investigates it clinical value in early detection of gastric cancer, providing a reasonably large cohort of patient data. However the presentation of data is poor and scientific writing in English need largely improved. Please amend the manuscript by addressing the following comments.

Response:Thank you for your positive comments. We have improved the presentation of data in Figure and Table sections according to your suggestion. In addition, we have carefully revised the manuscript and further improved the English language with the help from International Science Editing (https://ise.compuscript.ie/ise/chinese/certificate_download.php). The 16-digit verification code is 44749-R3JG84BCVO. We also have uploaded the International Science Editing Service Certification in the revised version. From the help of International Science Editing, the title has been revised as “Establishment of an absolute quantitative method to detect a plasma tRNA-derived fragment and its application in the non-invasive diagnosis of gastric cancer”.

Minor Comment 1: The description on miRNA in the first paragraph is not very related the context. Authors could introduce this later or in discussion part.

Response:Thank you for your constructive suggestion. We have deleted the description on miRNA in the first paragraph of “Introduction” section. 

Minor Comment 2: Suggest to include more information on tRFs, specially how they are regulating cell signaling pathway in second or third paragraph.

Response:Thank you for your constructive comments. We have included more information on tRFs, specially how they are regulating cell signaling pathway in second paragraph of “Introduction” section and in first paragraph of “Discussion” section. (Added reference No. 19 and No.25)

Minor Comment 3: Figure 1 should be more concise

Response:Thank you for the constructive suggestion. We have revised Figure 1 (Flow chart of the study design) and made it more concise.   

Minor Comment 4:In the method and result part, the authors seem to mix up when describing. For example Table 1 is mentioned in method part, but results are discussed very late in result part. I would suggest authors to re-write them all while keeping concise. Also author could consider to put assay set up in the main figures instead of supplemental figure. 

Response:Thank you for your constructive suggestions. We have rearranged Figure and Table, re-wrote them to keep concise. In addition, we have put assay set up in the main figures instead of supplemental figure. (Figure 2 and Figure 3 in "Results" section).

Minor Comment 5:For result part, authors should describe patient clearly (for example, 106 patients in table1 are not clear, are they from same cohort?).  

Response:Thank you for your constructive comments. We have carefully checked our manuscript and described patient cohort more clearly. We detected the expression level of tRF-27 in 106 preoperative plasma, and the relationship between the plasma tRF-27 levels and the clinicopathological factors of preoperative gastric cancer patients was analyzed in table1. We have re-wrote them to describe patient cohort clearly. (Page 7, "Results" section, 3.6. Plasma tRF-27 levels are closely related to tumor size and Ki67 expression in preoperative gastric cancer patients).        

Minor Comment 6:Explain results in more details: in result 3.1 and others, Please describe data by mentioning the average, mean, etc. Same with figure legend, author should clearly describe statistics terminology.

Response:Thank you for your constructive comments. We have explained results in more details: in result 3.1 (revised as result 3.3) and others, same with figure legend, we have added more statistics terminology.     

Such as in result 3.3, Plasma tRF-27 levels in healthy individuals, patients with benign gastric lesions, and patients with malignant gastric lesions. Data are presented as medians (interquartile ranges from 25% to 75%): healthy individuals (n = 120), 2.57 × 105 copies/mL (1.76 × 105 copies/mL–4.81 × 105 copies/mL); BL, patients with benign lesions (n = 48), 1.35 × 106 copies/mL (5.22 × 105 copies/mL–4.31 × 106 copies/mL); PL, patients with precancerous lesions (n = 48), 2.02 × 106 copies/mL (6.43 × 105 copies/mL–4.70 × 106 copies/mL); EC, patients with early cancer (n = 72), 2.15 × 106 copies/mL (5.09 × 105 copies/mL–9.36 × 106 copies/mL). The colors indicate different groups. ***, P < 0.001.

Such as in result 3.4, Plasma tRF-27 levels of patients with preoperative and postoperative gastric cancer and healthy individuals. Data are presented as medians (interquartile ranges from 25% to 75%). (A) Training set. Patients with preoperative gastric cancer (n = 48), 6.48 × 105 copies/mL (2.40 × 105–1.88 × 106); Patients with postoperative gastric cancer (n = 48), 5.00 × 105 copies/mL (1.24 × 105 copies/mL–1.95 × 106 copies/mL); Healthy individuals (n = 48), 2.11 × 105 copies/mL (1.67 × 105 copies/mL–3.12 × 105 copies/mL). (B) Validation set. Patients with preoperative gastric cancer (n = 58), 2.08 × 106 copies/mL (5.76 × 105 copies/mL–3.62 × 106 copies/mL); Patients with postoperative gastric cancer (n = 58), 7.44 × 105 copies/mL (1.46 × 105 copies/mL–1.75 × 106 copies/mL); Healthy individuals (n = 72), 3.48 × 105 copies/mL (1.95 × 105 copies/mL–5.74 × 105 copies/mL). (C) Plasma tRF-27 levels in postoperative patients (n = 71) were decreased. The colors indicate different groups. **, P < 0.01; ***, P < 0.001.

Minor Comment 7:In Figure 5, what is the cutoff for low and high expression of tRF-27? what is the criterion to set up in this way?  

Response:Thank you for your constructive comments. Figure 5 revised as Figure 7, the cutoff for low and high expression of tRF-27 was 724807 copies/mL (according to the Figure 6, the cutoff value was 724807 copies/mL). The criterion to set up in this way according to Youden's index. Youden's index: It is a method to evaluate the authenticity of screening test. Youden's index= (sensitivity + specificity) - 1, the larger the index, the better the effect and authenticity of the screening experiment.

Minor Comment 8:Authors should their research (background, findings, significance) in a broader way in discussion part

Response:Thank you for your constructive comments. We have enriched our research (background, findings, significance) in a broader way in discussion part. (Page 10, "Discussion" section, the first and second paragraph, "Conclusion" section, the first paragraph). 

Reviewer 2 Report

Yu., et al designed a quantitative detection method to count copy number of tRNA-derived fragment-27 existing in blood plasma, with a proposal of potential clinical utility to develop tRF-27 as a diagnostic biomarker for early gastric cancer. The novelty part resides in the design of primers and probes for rt-qPCR assay, followed by diagnostic value assessment with training and validation sets.  They also explored prognostic value with survival analysis.

Locations of figures and tables: I assume this is not the author’s fault/responsibility, but all figures should be shown on the same page or following pager when they are mentioned.

Abstract:

1.    As an assay development, the assay specific is not provided. Limit of detection (how many copies/ml)? Input amount?....

2.    Number of the sample sizes for each test group is missing.

3.    Diagnostic cutoff? Prognosis cutoff?

Introduction:

Paragraph 1 unfolded the unmet clinical challenge; however, the mechanism part is not directly relevant to this research goal.

Methods: How is the blood sample collected and transferred? Any specific tube types? Storage condition? How RNA prepared? And QC on input material?

Figure 1 is not mentioned/explained in the main text.

Table 1: What is “High (%)” and “Low (%)”? Missing from table legend. How is that defined?

Results:

3.1 There should be a clear description of Figure 2 here: sample size, median or average? SDT? Color designation? And before this, there should be a section describing the analytics of this method. As of 1. We developed a ……method with….primer…The limit of detection is…….; 2. With this method, we compared…….

3.2 Detail description of Figure 3 is missing. Figure legend of Fig 3 needs more information. What are the criteria used to draw this cutoff? What is the definition of early cancer? By tumor size?

3.3 How are the training set and validation set defined? Why is the validation set, the preoperative tRF-27 level (~20? by eye) significantly higher than that of the training set (~ 5)? Does the validation set include more cases with advanced diseases? Or higher clinicopathological factors?

3.5 What is the cutoff of the lower tRF and higher tRF groups? Need to be clearly stated here!!!

I would recommend a major revision. The result part needs a more careful description of each study design and figure presentation.

Author Response

Ref.: Ms. No. ijms-2033529

Title: "Establishment of an absolute quantitative method to detect a plasma tRNA-derived fragment and its application in the non-invasive diagnosis of gastric cancer"

Dear Reviewer,

On behalf of my co-authors, we thank you very much for giving us an opportunity to revise our manuscript entitled “Establishment of an absolute quantitative method to detect a plasma tRNA-derived fragment and its application in the non-invasive diagnosis of gastric cancer” (Manuscript ID: ijms-2033529). We appreciate the editors and reviewers very much for your positive and constructive comments. Those comments are all valuable and immensely helpful for improving our paper.    

We have studied all peer reviewers’ comments carefully and have addressed the suggestions raised by reviewers and editors. You endorsed this manuscript and we carefully revised the paper. More data and discussion have been added. We have submitted our documents with tracked changes to highlight the revisions. Point-to-point responses to the associate editors’ and reviewers’ comments are listed below this letter. Attached please find the revised version, which we would like to submit for your kind consideration.    

This manuscript was suitable for the special topic “Epigenetic Biomarkers and Applications for Liquid Biopsy Based Diagnostics”. We are willing to address your concerns carefully to make this manuscript suitable for publication, for the first author Xiuchong Yu is expected to graduate from the doctoral program in June 2023. In order to successfully obtain the doctoral degree, this paper needs to be published before March 2023. Every step forward, we are full of gladness and expectation.

We would like to express our great appreciation to you and reviewers for comments on our paper. Looking forward to hearing from you.

With best wishes,

Yours sincerely,

Junming Guo

Director and Professor

Department of Biochemistry and Molecular Biology,

and Zhejiang Key Laboratory of Pathophysiology,

School of Medicine, Ningbo University,

Ningbo 315211,

China

Tel: +86-574-87600756; Fax: +86-574-87608638;

Respond to Reviewer 2

General Comments: Yu., et al designed a quantitative detection method to count copy number of tRNA-derived fragment-27 existing in blood plasma, with a proposal of potential clinical utility to develop tRF-27 as a diagnostic biomarker for early gastric cancer. The novelty part resides in the design of primers and probes for RT-qPCR assay, followed by diagnostic value assessment with training and validation sets.  They also explored prognostic value with survival analysis.

Response:Thank you for your positive comments.

Minor Comment 1: Locations of figures and tables: I assume this is not the author’s fault/responsibility, but all figures should be shown on the same page or following pager when they are mentioned.

Response:Thank you for your positive comments. We have carefully rearranged Figure and Table, Figures have been shown following pager when they are mentioned.

Minor Comment 2: Abstract section: As an assay development, the assay specific is not provided. Limit of detection (how many copies/ml)? Input amount?....

Response:Thank you for your constructive comments.

As an assay development, this assay specific is non-invasive diagnosis of gastric cancer. The specific stem-loop structure reverse transcription primer, TaqMan probe, and amplification primers for tRF-27 were first designed and an absolute quantitative method was used to detect plasma tRF-27 level. We have added these sentences in “Abstract” section.

We have provided the detailed limit of detection and input amount in “Materials and methods” section. (Page 3-4, "Materials and methods" section). Details are as follows.    

2.1. Design of a specific stem-loop-structure reverse transcription primer of tRF-27.

We designed a specific stem-loop-structure reverse transcription primer to increase the length of tRF-27 for preparing cDNA (Supplementary Figure 1). The optimal concentration of the reverse transcription primer was 0.005 μM, that is, the amount added to the system was 2 μL. The total RNA amount added to the system was 6 μL. The reverse transcription reaction components were prepared as shown in Supplementary Table 1.  

Supplementary Table 1. Reverse transcription reaction system

Component

Volume (μL)

Total RNA

6

5×Polestar RT MasterMix (with dsDNase)

4

Stem-loop structure reverse transcription primer for tRF-27 (0.005μM)

2

RNase Free H2O

8

Total volume

Up to 20

2.2. Design of a TaqMan probe and amplification primers for qRT-PCR detection.

The preparation of the TaqMan probe was specific and accurate. The optimal concentration of the upstream primer, downstream primer, and TaqMan probe was set as 10 μM, that is, the amounts added to the system were 1.4 μL, 1.4 μL, and 0.8 μL, respectively. The cDNA amount added to the system was 0.8 μL. The set-up of the TaqMan probe reaction system is shown in Supplementary Table 3.

Supplementary Table 3. TaqMan probe reaction system

Component

Volume (μL)

2×5G qPCR PreMix (QPT-200, Toyobo, Japan)

10

Forward primer (10 μM)

1.4

Reverse primer (10 μM)

1.4

TaqMan Probe (10 μM)

0.8

cDNA template

0.8

ddH2O

5.6

Total volume

20

2.4. Detection of tRF-27 levels in patients with benign and malignant gastric lesions

Peripheral venous blood was collected into ethylenediamine tetraacetic acid anticoagulation tubes and centrifuged for 10 min at 1500 g and 4 °C. The upper plasma (250 μL) was transferred for extracting total RNA with TRIzol LS (Invitrogen, Carlsbad, CA, USA). RNA was dissolved in 7 μL of RNase-free water, and used 1μL to measure the purity and concentration with a Nanodrop UV spectrophotometer. If the ratio of A260-to-280 was between 1.8 and 2.1, the RNA purity met the requirements for further experimentation. Then the remaining 6 μL RNA was used for reverse transcription.

Minor Comment 3: Abstract section: Number of the sample sizes for each test group is missing.

Response:Thank you for your constructive suggestion. We have added number of the sample sizes for each test group in “Methods” of “Abstract” section. (Page 1, "Abstract" section).  

Minor Comment 4: Abstract section: Diagnostic cutoff? Prognosis cutoff?

Response:Thank you for your constructive suggestion. Plasma tRF-27 levels were significantly increased in gastric cancer, and the area under the receiver operating characteristic curve was 0.7767, when the cutoff value was 724807 copies/mL, with a sensitivity and specificity 0.6226 and 0.8917, respectively. The positive predictive and negative predictive values were 83.50% and 72.80%, respectively. We have clarified and added the cutoff value in “Results” of “Abstract” section.

Minor Comment 5: Introduction section: Paragraph 1 unfolded the unmet clinical challenge; however, the mechanism part is not directly relevant to this research goal.

Response:Thank you for your constructive suggestion. We have deleted the mechanism part which not directly relevant to this research goal in the first paragraph. (Page 1, "Introduction" section, the first paragraph).  

Minor Comment 6:Methods section: How is the blood sample collected and transferred? Any specific tube types? Storage condition? How RNA prepared? And QC on input material?

Response:Thank you for your constructive suggestions. Peripheral venous blood was collected into ethylenediamine tetraacetic acid anticoagulation tubes and centrifuged for 10 min at 1500 g and 4 °C. The upper plasma (250 μL) was transferred for extracting total RNA with TRIzol LS (Invitrogen, Carlsbad, CA, USA). RNA was dissolved in 7 μL of RNase-free water, and used 1μL to measure the purity and concentration with a Nanodrop UV spectrophotometer. If the ratio of A260-to-280 was between 1.8 and 2.1, the RNA purity met the requirements for further experimentation. Then the remaining 6 μL RNA was used for reverse transcription. The absolute quantitation method was used to measure plasma tRF-27 levels in plasma.  

Minor Comment 7:Results section: Figure 1 is not mentioned/explained in the main text. Table 1: What is “High (%)” and “Low (%)”? Missing from table legend. How is that defined?

Response:Thank you for the constructive suggestion. We have mentioned/explained Figure 1 (Flow chart of the study design) in the main text, the last paragraph of “Introduction” section and the first paragraph of “Discussion” section.   

We have defined the group “High (%) and Low (%)” in table legend. the cutoff for low and high expression of tRF-27 was 724807 copies/mL (according to Figure 6, the cutoff value was 724807 copies/mL). The criterion to set up in this way according to Youden's index. Youden's index: It is a method to evaluate the authenticity of screening test. Youden's index= (sensitivity + specificity) - 1, the larger the index, the better the effect and authenticity of the screening experiment. In the preoperative gastric cancer patient group, there were 66 patients with tRF-27 expression levels greater than or equal to 724807 copies/mL and 40 patients with tRF-27 expression level lower than 724807 copies/mL.

Minor Comment 8:Results section: 3.1 There should be a clear description of Figure 2 here: sample size, median or average? SDT? Color designation? And before this, there should be a section describing the analytics of this method. As of 1. We developed a ……method with….primer…The limit of detection is…….; 2. With this method, we compared…….

Response:Thank you for your constructive comments. We have described Figure 2 (revised as Figure 4) more clearly. Plasma tRF-27 levels in healthy individuals, patients with benign gastric lesions, and patients with malignant gastric lesions. Data are presented as medians (interquartile ranges from 25% to 75%): healthy individuals (n = 120), 2.57 × 105 copies/mL (1.76 × 105 copies/mL–4.81 × 105 copies/mL); BL, patients with benign lesions (n = 48), 1.35 × 106 copies/mL (5.22 × 105 copies/mL–4.31 × 106 copies/mL); PL, patients with precancerous lesions (n = 48), 2.02 × 106 copies/mL (6.43 × 105 copies/mL–4.70 × 106 copies/mL); EC, patients with early cancer (n = 72), 2.15 × 106 copies/mL (5.09 × 105 copies/mL–9.36 × 106 copies/mL). The colors indicate different groups. ***, P < 0.001.

And before this, We have added more result to describe the analytics of this method in result 3.1, result 3.2. We developed a new absolute quantitative method to detect tRF-27 in the plasma. Successful design of tRF-27-specific reverse transcription primer, TaqMan probe, and amplifi-cation primerswith specific reverse transcription primer, TaqMan probe and amplification primers. With this new absolute quantitative method, we compared tRF-27 copy number in plasma.  

Minor Comment 9:Results section: 3.2 Detail description of Figure 3 is missing. Figure legend of Fig 3 needs more information. What are the criteria used to draw this cutoff? What is the definition of early cancer? By tumor size?

Response: According to your suggestion, we have added more information to Figure 3 (revised as Figure 6) and Figure legend. The criteria used to draw the cutoff was according to Youden's index. Youden's index: It is a method to evaluate the authenticity of screening test. Youden's index= (sensitivity + specificity) - 1, the larger the index, the better the effect and authenticity of the screening experiment. According to American Joint Committee on Cancer (AJCC), the definition of early cancer: Limited to mucosa or submucosa, regardless of lymph node metastasis.

Minor Comment 10:Results section: 3.3 How are the training set and validation set defined? Why is the validation set, the preoperative tRF-27 level (~20? by eye) significantly higher than that of the training set (~ 5)? Does the validation set include more cases with advanced diseases? Or higher clinicopathological factors?

Response:Thank you for your constructive comments. The result 3.3 have been revised as result 3.4. The training set samples were collected in 2018, The validation set samples were collected in 2019. Data are presented as medians (interquartile ranges from 25% to 75%), training set preoperative (n=48), 6.48×105 copies/mL (2.40×105 copies/mL-1.88×106 copies/mL); validation set preoperative (n=58), 2.08×106 copies/mL (5.76×105 copies/mL-3.62×106 copies/mL); The validation set include more cases.

Minor Comment 11:Results section: 3.5 What is the cutoff of the lower tRF and higher tRF groups? Need to be clearly stated here.

Response:Thank you for your constructive comments. Results section: 3.5 revised as Results section: 3.7. The cutoff for low and high expression of tRF-27 was 724807 copies/mL (according to the Figure 6, the cutoff value was 724807 copies/mL). In the preoperative gastric cancer patient group, there were 66 patients with tRF-27 expression levels greater than or equal to 724807 copies/mL and 40 patients with tRF-27 expression level lower than 724807 copies/mL. We have clearly stated in Results section: 3.5 and in the Figure 7 legend.

Round 2

Reviewer 1 Report

Dear Authors,

The manuscript quality has been largely improved.

Please indicate in the text: 1) the qPCR machine and software you are using 2)and how you calculate the copy number(software? formula?) 3) also how area under the curve (AUC), cutoff value, sensitivity, specificity, positive predictive value, and negative predictive value are analysized in detail (software?)

Author Response

Ref.: Ms. No. ijms-2033529

Title: "Establishment of an absolute quantitative method to detect a plasma tRNA-derived fragment and its application in the non-invasive diagnosis of gastric cancer"

Dear Reviewer,

On behalf of my co-authors, we thank you very much for giving us an opportunity to revise our manuscript entitled “Establishment of an absolute quantitative method to detect a plasma tRNA-derived fragment and its application in the non-invasive diagnosis of gastric cancer” (Manuscript ID: ijms-2033529). We appreciate you very much for your positive and constructive comments. Those comments are all valuable and immensely helpful for improving our paper.    

We have studied your comments carefully and have addressed the suggestions raised by you. You endorsed this manuscript and we carefully revised the paper. We have submitted our documents with tracked changes to highlight the revisions. Point-to-point responses to the comments are listed below this letter. Attached please find the revised version, which we would like to submit for your kind consideration.   

This manuscript was suitable for the special topic “Epigenetic Biomarkers and Applications for Liquid Biopsy Based Diagnostics”. We are willing to address your concerns carefully to make this manuscript suitable for publication, for the first author Xiuchong Yu is expected to graduate from the doctoral program in June 2023. In order to successfully obtain the doctoral degree, this paper needs to be published before March 2023. Every step forward, we are full of gladness and expectation.

We would like to express our great appreciation to you for comments on our paper. Looking forward to hearing from you.

With best wishes,

Yours sincerely,

Junming Guo

Director and Professor

Department of Biochemistry and Molecular Biology,

and Zhejiang Key Laboratory of Pathophysiology,

School of Medicine, Ningbo University,

Ningbo 315211,

China

Tel: +86-574-87600756; Fax: +86-574-87608638;

Respond to Reviewer 1

General Comments: The manuscript quality has been largely improved.

Response:Thank you for your positive comments.

Minor Comment 1: Please indicate in the text the qPCR machine and software you are using.

Response:Thank you for your constructive suggestion. We have indicated in the text the qPCR machine and software we used. The Applied Biosystems™ QuantStudio™ 3 Quantitative Real-Time PCR System (Thermo Fisher Scientific, Waltham, MA, USA) coupled to QuantStudio™ Design and Analysis Software was used to detect the Cq value. (Page 4, “Materials and methodssection, 2.3. Establishment of an absolute quantitative method to measure plasma tRF-27 levels).

Minor Comment 2: Please indicate in the text how you calculate the copy number (software? formula?)

Response:Thank you for your constructive comments. We have indicated in the text how we calculated the copy number. Based on the copy number of the template and the Cq value of qRT-PCR, a standard curve of the tRF-27 copy number was generated. According to the Cq value and the standard curve formula, the absolute amount of tRF-27 in plasma was calculated. (Page 4, Materials and methods” section, 2.3. Establishment of an absolute quantitative method to detect plasma tRF-27 levels). 

Graphs were plotted with GraphPad Prism 9 Software. (Page 5, Materials and methods” section, 2.8. Statistical analysis). 

Minor Comment 3: Please indicate in the text how area under the curve (AUC), cutoff value, sensitivity, specificity, positive predictive value, and negative predictive value are analysized in detail (software?)

Response:Thank you for the constructive suggestion. We have indicated in the text how area under the curve (AUC), cutoff value, sensitivity, specificity, positive predictive value, and negative predictive value are analysized in detail. The area under the curve (AUC), cutoff value, sensitivity, specificity, positive predictive value, and negative predictive value were analyzed by GraphPad Prism 9 Software (version 9.0.0, San Diego, CA, USA). The ROC curve was drawn with 1-specificity as the abscissa and sensitivity as the ordinate. The size of the AUC can quantitatively and specifically indicate the accuracy of the diagnostic test. The criterion of the cutoff value was set up according to Youden’s index. Youden’s index is a method to evaluate the authenticity of the screening test. Youden’s index was calculated as follows: = (sensitivity + specificity) ̵ 1, and the larger the index, the better the effect and the authenticity of the screening test. The positive predictive value was calculated as follows: = true positive / (true positive + false positive). The negative predictive value was calculated as follows: = true negative / (true negative + false negative). (Page 5, “Materials and methods” section, 2.6. Construction of a receiver operating characteristic curve to evaluate the diagnostic value of tRF-27 in gastric cancer). 

Statistical analysis was performed with Statistical Product and Service Solutions (SPSS) Software (version 23.0, Chicago, IL, USA). Graphs were plotted with GraphPad Prism 9 Software. (Page 5, “Materials and methods” section, 2.8. Statistical analysis). 

Reviewer 2 Report

Great improvement. No major questions are unresolved. Minor language issues need to be crafted.

Author Response

Ref.: Ms. No. ijms-2033529

Title: "Establishment of an absolute quantitative method to detect a plasma tRNA-derived fragment and its application in the non-invasive diagnosis of gastric cancer"

Dear Reviewer,

On behalf of my co-authors, we thank you very much for giving us an opportunity to revise our manuscript entitled “Establishment of an absolute quantitative method to detect a plasma tRNA-derived fragment and its application in the non-invasive diagnosis of gastric cancer” (Manuscript ID: ijms-2033529). We appreciate you very much for your positive and constructive comments. Those comments are all valuable and immensely helpful for improving our paper.    

We have studied your comments carefully and have addressed the suggestions raised by you. You endorsed this manuscript and we carefully revised the paper. We have submitted our documents with tracked changes to highlight the revisions. Point-to-point responses to the comments are listed below this letter. Attached please find the revised version, which we would like to submit for your kind consideration.   

This manuscript was suitable for the special topic “Epigenetic Biomarkers and Applications for Liquid Biopsy Based Diagnostics”. We are willing to address your concerns carefully to make this manuscript suitable for publication, for the first author Xiuchong Yu is expected to graduate from the doctoral program in June 2023. In order to successfully obtain the doctoral degree, this paper needs to be published before March 2023. Every step forward, we are full of gladness and expectation.

We would like to express our great appreciation to you for comments on our paper. Looking forward to hearing from you.

With best wishes,

Yours sincerely,

Junming Guo

Director and Professor

Department of Biochemistry and Molecular Biology,

and Zhejiang Key Laboratory of Pathophysiology,

School of Medicine, Ningbo University,

Ningbo 315211,

China

Tel: +86-574-87600756; Fax: +86-574-87608638;

Respond to Reviewer 2

General Comments: Great improvement. No major questions are unresolved. Minor language issues need to be crafted.

Response:Thank you for your positive comments. We have carefully revised the manuscript and further improved the English language again with the help from International Science Editing (https://ise.compuscript.ie/ise/admin/certificate_download.php). The 16-digit verification code is 44749-XSICAQOUZ1. We also have uploaded the International Science Editing Service Certification in Supplementary materials of the revised version. 
